# Structure of the active pharmaceutical ingredient bismuth subsalicylate

Erik Svensson Grape[1], Victoria Rooth[1], Mathias Nero [1], Tom Willhammar [1✉] & A. Ken Inge [1✉]

Structure determination of pharmaceutical compounds is invaluable for drug development but remains challenging for those that form as small crystals with defects. Bismuth subsalicylate, among the most commercially significant bismuth compounds, is an active ingredient in over-the-counter medications such as Pepto-Bismol, used to treat dyspepsia and *H. pylori* infections. Despite its century-long history, the structure of bismuth subsalicylate is still under debate. Here we show that advanced electron microscopy techniques, namely three-dimensional electron diffraction and scanning transmission electron microscopy, can give insight into the structure of active pharmaceutical ingredients that are difficult to characterize using conventional methods due to their small size or intricate structural features. Hierarchical clustering analysis of three-dimensional electron diffraction data from ordered crystals of bismuth subsalicylate revealed a layered structure. A detailed investigation using high-resolution scanning transmission electron microscopy showed variations in the stacking of layers, the presence of which has likely hindered structure solution by other means. Together, these modern electron crystallography techniques provide a toolbox for structure determination of active pharmaceutical ingredients and drug discovery, demonstrated by this study of bismuth subsalicylate.

[1] Department of Materials and Environmental Chemistry, Stockholm University, 10691 Stockholm, Sweden. ✉email: tom.willhammar@mmk.su.se; andrew.inge@mmk.su.se

The physical, chemical and therapeutic properties of active pharmaceutical ingredients (APIs) are governed by their molecular structures and intermolecular interactions. Many APIs are crystalline substances, with periodic arrangements of their constituent molecules or ions. The specific arrangement of molecules and their intermolecular interactions affect stability and solubility, which in turn influences bioavailability, efficacy, and dosage. Therefore, determining the structures of pharmaceutical compounds is an integral part of drug formulation. Traditionally the method of choice for crystal structure determination has been single-crystal X-ray diffraction (SCXRD). However, the technique requires large specimens and is not readily applicable for submicrometer-sized crystals. While structure determination of small crystallites can often be performed by powder X-ray diffraction (PXRD), the technique can at times struggle with complicated and disordered structures.

These reasons among others have previously prevented the structure determination of the API bismuth subsalicylate (BSS), a crystalline compound made from $Bi^{3+}$ cations, $O^{2-}$ anions, and salicylate anions (Hsal⁻, Fig. 1a). It is administered in its crystalline state and is the API of popular over-the-counter medications such as Pepto-Bismol, commonly used to treat gastrointestinal disorders such as dyspepsia and diarrhoea. Numerous studies have confirmed the efficacy of BSS as an antimicrobial, anti-inflammatory, and antacid agent[1–3]. Antimicrobial properties of several other bismuth compounds such as bismuth subgallate, bismuth subnitrate, and colloidal bismuth tartrate have been reported[4], and specifically, it has been demonstrated that colloidal bismuth subcitrate and ranitidine bismuth citrate can even combat antibiotic resistance in bacteria[5] and suppress SARS-CoV-2 replication[6].

Formulations of BSS were developed in 1900 to treat *Campylobacter* infections, a major cause of infant deaths at the time[7]. Since the discovery in the 1980s by Nobel laureates Barry Marshal and Robin Warren[8] of *Helicobacter pylori*, a bacterium harboured by 60% of the global population, bismuth compounds, including BSS, have received renewed interest by effectively treating peptic ulcer disease[9]. In 1990, a report from Procter & Gamble (P&G) estimated that over 10 billion doses of Pepto-Bismol had been consumed and that it was found in approximately 60% of U.S. households[7]. In 2019, overall sales of >20 million units grossed over $100 million in the U.S. alone, making it the most sold stomach remedy in the country[10]. Despite its century-long history and continuing widespread use, the structure of BSS has remained unknown and only a limited understanding has been established of its mechanisms of action.

Speculations on the formula and structure of BSS have been published in many chemical and pharmaceutical databases, websites, patents, textbooks, and articles, where BSS is often represented as a simple metal complex. Although efforts have been made to determine its crystal structure, obtaining sufficiently large specimens of BSS for SCXRD has not been possible —likely due to its poor solubility in water. Due to the difficulties in characterizing BSS, several model bismuth compounds have been developed through various approaches, including the synthesis of bismuth thiosalicylates[11], the incorporation of water or organic solvent molecules into the crystal structures[12–15], or by altering the Bi:Hsal stoichiometry[16,17], which is 1:1 for BSS.

Three-dimensional electron diffraction (3DED)[18] techniques, such as cRED, ADT, fast-EDT, and MicroED, can be applied to obtain single-crystal diffraction data on submicrometer crystallites for the determination of their average ordered structures. This has been facilitated through the development of methodology and hardware in recent years[19–24] and has allowed for faster and higher quality measurements for the structure determination of proteins[20,22,25], inorganics[26], and organics[27,28], including pharmaceuticals[29,30] as well as various bismuth compounds[31–34]. Concurrently, aberration-corrected high-resolution scanning transmission electron microscopy (STEM) has evolved into an essential technique for atomic-scale structural investigations, particularly of local disorder. The recent development of STEM techniques such as integrated differential phase contrast (iDPC) has allowed for studies on beam-sensitive specimens, including studies of organic molecules inside inorganic framework materials[35,36].

In this work, we uncover the structure of BSS by applying these modern transmission electron microscopy (TEM) techniques. Screening of various sources of BSS reveals differing degrees of long-range order within crystals. 3DED data on BSS crystals with a high degree of long-range order unveils BSS as a layered material, while STEM imaging of crystals with less order reveals the presence of disordered domains due to various stacking sequences of the layers.

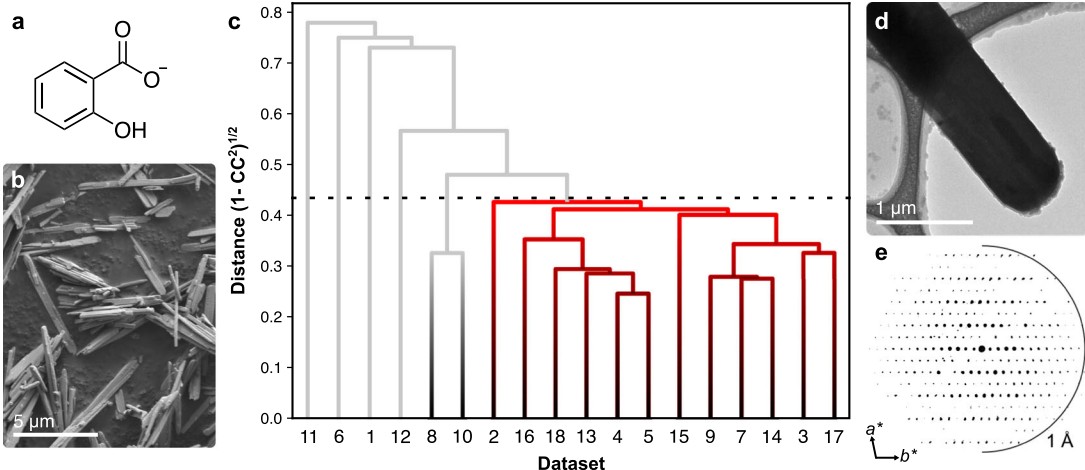

**Fig. 1 Three-dimensional electron diffraction studies on BSS. a** Molecular structure of the salicylate anion (Hsal⁻). **b** Scanning electron micrograph of bismuth subsalicylate from Sigma-Aldrich (BSS-SA). **c** A dendrogram for the hierarchical clustering of the 18 individual 3DED data sets with the correlation coefficient cut-off shown as a dashed line. The red branch represents 12 highly correlated data sets with a distance below 0.44, which were merged into a single combined 3DED data set. **d** TEM image of one of 18 BSS-SA crystals used for the collection of 3DED data. **e** The reconstructed reciprocal space projection of a single 3DED data set viewed along c* (half-circle is drawn at a resolution of 1 Å).

## Results

To identify appropriate BSS crystals for detailed investigations, five samples from different suppliers or formulations were screened by PXRD (Supplementary Fig. 1). All samples were crystalline and had characteristic PXRD patterns of BSS. The differences observed in the powder patterns are attributed to either the presence of crystalline inactive ingredients or various degrees of disorder within the materials. Based on the quality of the PXRD patterns, phase purity and commercial significance, our investigation narrowed its focus to two samples: BSS purchased from the chemical provider Sigma-Aldrich (BSS-SA) and BSS isolated from Pepto-Bismol original liquid formulation (BSS-PB).

Inspection of both BSS-SA and BSS-PB samples by scanning electron microscopy (SEM) and TEM imaging revealed crystals with a plank-shaped morphology that appeared homogeneous with no obvious indications of impurities (Fig. 1b, d and Supplementary Fig. 2). Despite efforts to elucidate the structure by synchrotron SCXRD, the crystals proved too small and agglomerated. Structure determination by PXRD was not successful due to the preferred orientation of the crystals causing both over and underemphasized intensities, but also due to diffuse scattering and a rather complicated crystal structure, as disclosed by 3DED experiments (Fig. 1e).

**Structure determination of BSS.** 3DED data sets were collected at 98 K on crystals from BSS-PB, which had been centrifuged out of suspension and washed with water. The quality of data sets suffered from a variety of problems including inadequate data resolution, irregular peak shapes, twinning, and in some cases diffuse scattering (Supplementary Fig. 3). A few of the data sets could be indexed to a triclinic unit cell, however, no reasonable structure model was obtained. Higher quality 3DED data were acquired from crystallites of BSS-SA (Supplementary Fig. 4). Data sets from 18 crystallites could be indexed with a triclinic unit cell ($a = 8.35$ Å, $b = 12.17$ Å, $c = 18.09$ Å, $\alpha = 77.9°$, $\beta = 83.2°$, $\gamma = 76.7°$). Initial structure solution was attempted on individual 3DED data sets but was unsuccessful. This was attributed to the low completeness of the individual data sets (≤50%) caused by the low symmetry of the crystals and the limited tilt range intrinsic to the TEM. To improve data completeness, hierarchical clustering analysis (HCA) was performed to merge data sets that were most similar in terms of measured reflection intensities[37]. A distance metric based on the correlation coefficient (CC) of overlapping data was generated for all possible pairs of data sets. This resulted in two separate clusters with a CC of at least 0.90, corresponding to a distance metric of 0.44 (Fig. 1c). Structure solution by direct methods using the data of the large cluster, composed of 12 individual data sets with overall completeness of 84.6%, resulted in a model with all non-hydrogen atoms appropriately located in the crystal structure with the space group $P$-1 (Supplementary Table 1).

As revealed by 3DED, BSS proved to be a coordination polymer with a layered structure (Fig. 2a and Supplementary Fig. 5). The crystal structure and its asymmetric unit of $Bi_4O_4(Hsal)_4$ is in accordance with the commonly presented empirical formula of $BiC_7H_5O_4$, as well as with elemental and thermogravimetric analyses (Supplementary Fig. 6). Considering the stoichiometry of the material and the presence of $O^{2-}$ anions, the systematic name of BSS is bismuth(3+) oxide salicylate. In BSS, $Bi^{3+}$ cations are bridged by $O^{2-}$ anions into bismuth-oxo rods that extend along the $a$-axis and form the inorganic building unit (IBU) of the structure (Fig. 2a–c). Along the centre of the rods, $O^{2-}$ anions bridge alternatingly three ($\mu_3$) and four ($\mu_4$) $Bi^{3+}$ cations. The IBU of BSS is very similar but not identical to

those found in a number of previously reported bismuth-based coordination polymers[33,38–41]. There are two types of salicylate anions ($Hsal^-$) in the BSS structure. One type of $Hsal^-$ coordinates via the carboxylate group only to $Bi^{3+}$ cations of a single rod, while the phenolic group does not coordinate to any $Bi^{3+}$ cations. The other type also coordinates to $Bi^{3+}$ cations through the carboxylate group; however, the phenolic group coordinates to $Bi^{3+}$ cations in adjacent IBUs, similarly to the bridging $Hsal^-$ ligands in the $[Bi_4O_2(Hsal)_8]\cdot 2MeCN/MeNO_2$ structures,[14] and essentially link the rods along the $b$-axis into centrosymmetric layers in the $ab$-plane. These layers stack along the $c$-axis and weakly interact with one another via dispersion forces. As the unit cell is only one layer thick, IBUs in neighbouring layers are oriented in the same direction in the ordered crystal.

Deprotonation of the carboxylate groups is suggested by the 3DED data based on the generally shorter Bi-O(carboxylate) distances (2.3–3.1 Å) compared to the Bi-O(phenolic) distances (3.1–3.2 Å) (Supplementary Table 2). IR spectroscopy also indicates full deprotonation of the carboxylate groups (Supplementary Fig. 7). The protonation, as assigned in Fig. 2d, results in a charge-balanced material. Similar protonation assignment of the $Hsal^-$ ligands has been reported in structures such as $[Bi_4O_2(Hsal)_8]\cdot 2MeCN/MeNO_2$ and $[Bi(Hsal)_3(H_2O)]$[14,15].

Due to the poor quality of the 3DED data on BSS-PB crystals, PXRD data were instead utilized to investigate the structure of BSS-PB. Structure refinement of BSS-PB showed overall good agreement with the BSS structure obtained by 3DED on BSS-SA (Supplementary Fig. 8 and Supplementary Table 3). However, diffuse scattering and asymmetric peak shapes in the PXRD pattern (Supplementary Fig. 9) suggested the presence of disorder in the BSS-PB samples.

**High-resolution STEM imaging of BSS layers.** To further validate the structure of BSS-PB and investigate structural disorder, aberration-corrected STEM imaging was applied. Imaging was performed using both annular dark field (ADF) as well as iDPC signals. The ADF contrast scales rapidly with the atomic number, thus highlighting heavier elements such as Bi. The iDPC contrast, on the other hand, scales linearly with atomic number, thereby emphasizing lighter elements when compared to ADF[42].

Crystals of BSS-PB consisted of large ordered domains in projection consistent with the structure of BSS-SA determined by 3DED. Images along the [100] direction of BSS-PB revealed a similar orientation of the IBUs (Fig. 3). The ADF contrast shows well-resolved projected positions of the $Bi^{3+}$ ions of the IBU (Fig. 3a), whereas the iDPC contrast, in addition, showed enhanced contrast in locations consistent with the positions of the salicylate anions, although not as well resolved (Fig. 3b).

However, upon inspection of other sections of the BSS-PB crystals, iDPC-STEM images revealed different types of disorder, particularly an inconsistency in the stacking of layers (Fig. 4 and Supplementary Fig. 10). It can be seen that the orientation of the layers vary, which can be caused by a twofold rotation of the layers around the $b$-axis or perpendicular to the $ab$-plane. In some domains, it was evident that the unit cell was doubled along the $c$-axis and the unit cell angle $\alpha$ changed due to a periodic alternation of the layer orientation.

In other domains, the orientations of the layers appear to be random and the disorder is observed as diffuse features in the Fast Fourier transform (FFT) of the image, as shown in Fig. 4d–f. The fact that domains of disordered sequences are observed explains the occurrence of inconsistent peak shapes and diffuse scattering in the PXRD pattern, as well as the initial difficulties in obtaining a structure model. As such, the material appears to have

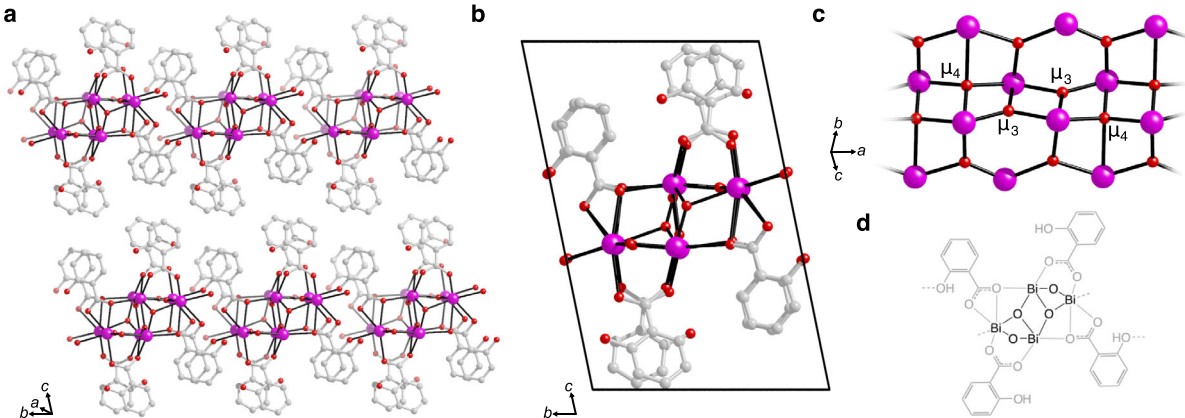

**Fig. 2 The crystal structure of bismuth subsalicylate. a** Stacking of the layers found in the BSS structure. Bi-O bonds are drawn as black lines. Hydrogen atoms have been omitted for clarity. **b** The unit cell of BSS viewed down the *a*-axis. **c** A section of the bismuth-oxo rod forming the inorganic building unit which includes $O^{2-}$ anions coordinated to alternately three and four $Bi^{3+}$ cations, $\mu_3$ and $\mu_4$, respectively. **d** A molecular sketch of a section of the BSS structure with bonds along the rod omitted for clarity.

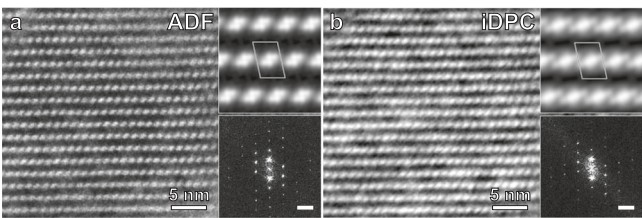

**Fig. 3 STEM images of a highly ordered section of a BSS-PB crystal along [100]. a** Annular dark field (ADF) scanning transmission electron microscopy (STEM) image. **b** Integrated differential phase contrast (iDPC) STEM image. Lattice-averaged maps with p2 symmetry imposed and fast Fourier transforms (scale bar is equal to $0.1 \text{ Å}^{-1}$) are shown as insets for each image.

(1) ordered domains with a *c*-axis of 17 Å with a single layer orientation (Fig. 4a, area 1, and 4g), (2) ordered domains with a doubled *c*-axis of 34 Å and alternating layer orientation for adjacent layers (Fig. 4a, area 3, and 4h), (3) domains of disordered stacking of the layers (Fig. 4a, area 2, and 4c), as well as (4) defects where the orientation changes within an individual layer (Fig. 4b).

## Discussion

Elucidation of the structure of commercial BSS provides a major step towards understanding the properties of one of the most commercially significant bismuth compounds. The fact that BSS is practically insoluble in water and the hydrophobic properties of the powder can be partly attributed to the continuous structure of the coordination polymer, where the less polar section of the salicylate anions form the outer surfaces of the layers, while all ionic and hydrophilic components, such as the phenolic group, carboxylate group, $\mu_3$-$O^{2-}$, $\mu_4$-$O^{2-}$, and $Bi^{3+}$, are contained within the layers. This hydrophobic character is also in alignment with the fact that BSS embedded in hydrophobic resin starts to dissolve from the (010) facets (Supplementary Fig. 11). In addition to its high stability in water, BSS also demonstrated decent stability in aqueous solutions of HCl (Supplementary Fig. 12). No changes were observed in the PXRD patterns of BSS treated at a pH of 3 or higher. At a pH of 2, a small proportion of BSS converted into bismuth oxychloride and was fully converted at a pH of 1, which is in line with previous indications that some of the administered BSS could even reach the small intestine[7,43]. Considering this, interactions between the hydrophobic exterior

of the BSS crystals and the gastric lining could to some extent govern the pharmacodynamics of this long-used formulation[44].

The various forms of analysis, including PXRD, 3DED, and STEM imaging, indicated that the two investigated samples, BSS-PB and BSS-SA, were built from the same layers but that the samples differed in terms of the degree of crystallinity and disorder in their structures. Understanding the differences in local ordering of BSS opens opportunities to develop analogues containing unique stacking sequences or higher degrees of exfoliation, which may influence the properties and the efficacy of the API. Considering this, the combination of electron crystallography tools used, 3DED and STEM, is expected to become an important part of drug discovery and structure determination of APIs.

## Methods

**Sourcing and synthesis of materials.** Commercially available Pepto-Bismol suspension manufactured by P&G was purchased from the supermarket Foodland Pukalani, HI, USA. Prior to performing 3DED and scanning transmission electron microscopy (STEM) measurements, 5 mL of the suspension was washed with 50 mL of water and centrifuged at $6000 \times g$ for 5 min whereafter it was left to dry overnight under ambient conditions (this washed powder is referred to as BSS-PB).

Commercially available bismuth subsalicylate (BSS, CAS: 14882-18-9) was purchased from Sigma-Aldrich (BSS-SA). The untreated powder was used for structure determination.

Laboratory-made bismuth subsalicylate was prepared by adding 72 mg bismuth(III) oxide (Aldrich, 99.999% trace metal basis) and 128 mg salicylic acid (Sigma-Aldrich, ACS reagent, ≥99%) to a 5 mL Pyrex tube filled with 3 mL of deionized water and sealed with a PTFE cap. The tube was heated to 140 °C while stirring at 800 rpm for 3 h. The solid was then filtered off, washed with 50 mL of hot deionized water (>95 °C), and left to dry in ambient conditions overnight.

**3DED measurements.** Specimen for 3DED studies were prepared by suspending BSS powder in deionized water, whereafter it was drop-cast onto a copper grid covered with a holey carbon film. The prepared grid was then put into a Gatan 914 cryo-transfer holder and cooled down to approximately 170 K before being inserted into the ultra-high vacuum of the TEM, reaching a final temperature of 96 K before data collection was started. 3DED data were collected using a JEOL JEM2100 TEM operating at 200 kV, equipped with a Timepix detector from Amsterdam Scientific Instruments, while continuously rotating the crystal at $0.45°\,s^{-1}$. The experiments were carried out using Instamatic[24], data analysis was performed using a hierarchical clustering procedure, discriminating on the correlation coefficient of the observed intensities for the collected data sets[37], and data reduction was performed by XDS[45]. For the clustering, a correlation coefficient cut-off of 0.90 was chosen for the merging of data sets. The acquired intensities were then used to solve the structure of BSS with SHELXS whereafter it was refined using SHELXL[46], using electron scattering factors extracted from SIR2014[47]. The subsequent refinement of the electron-diffraction model against high-resolution PXRD data on a sample of BSS from Pepto-Bismol (BSS-PB) was carried out in TOPAS-Academic V6[48], using a geometric approach to account for the anisotropic peak broadening in the data[49].

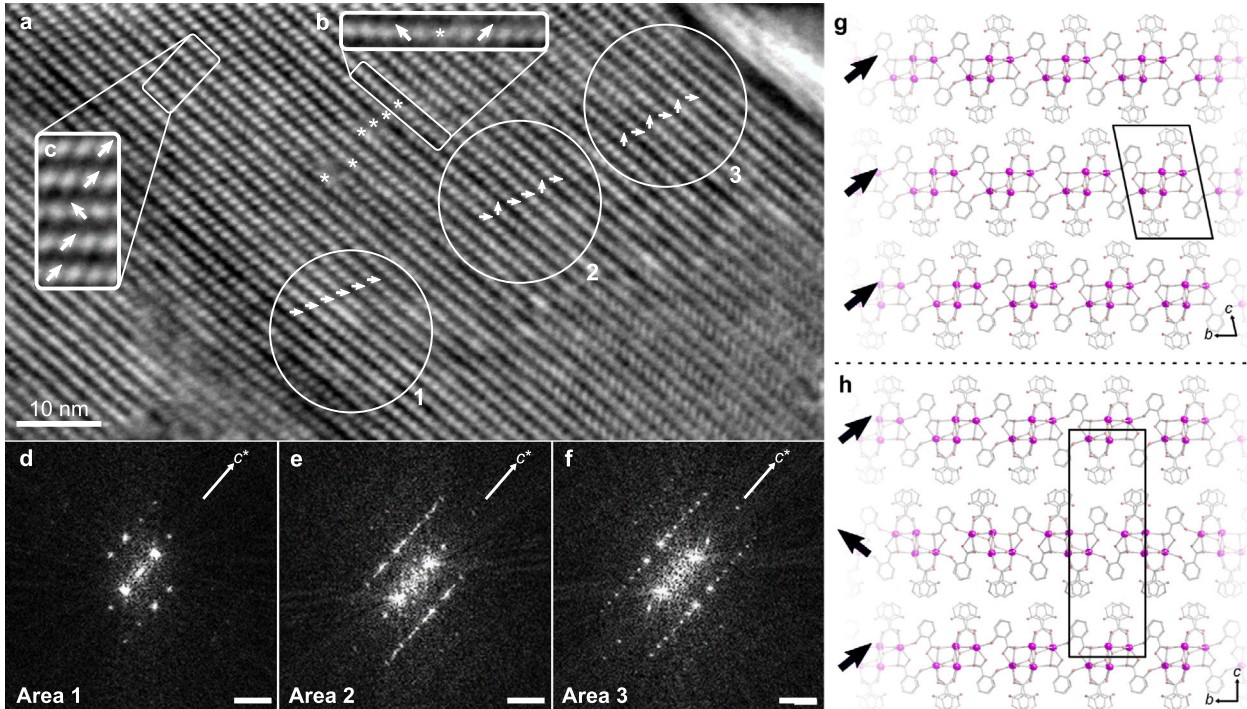

**Fig. 4 iDPC-STEM image reveals disorder in BSS-PB. a–c** Domains of disorder are found in BSS while imaged along the [100] direction, showing alternation of the layer orientation both within (**b**) and between the layers (**c**). Arrows indicate the orientation of the layers. Stars (\*) indicate switching of the orientation within a single layer. **d** Domains with a 17 Å $c$-axis are found as evidenced by the fast Fourier transform (FFT), showing a single layer orientation. **e** Domains of disordered stacking are also observed as diffuse streaking perpendicular to the layer. **f** Domains showing a doubled 34 Å $c$-axis, arising from an alternating orientation of the individual layers (scale bar in **d–f** corresponds to 0.1 Å$^{-1}$). **g** The structure of BSS as determined from 3DED data with all similar orientations of the layers. **h** A structure with alternating layer orientations, consistent with the doubled $c$-axis.

**Scanning transmission electron microscopy measurements**. In order to facilitate STEM imaging of crystalline BSS, a small amount (~20 μg) of the washed and dried powder from the Pepto-Bismol original liquid formulation (BSS-PB) was embedded in a resin (LR White) in a gelatin capsule (size 00) and then hardened at 60 °C for 24 h. Ultra-thin sectioning, with an estimated section thickness of 40 nm, was later carried out using a Leica Ultracut UCT with a 45° diamond knife from Diatome. The sections were then transferred to carbon-coated copper grids (EMS-CF150-Cu-UL).

STEM images of BSS-PB were obtained using a Thermo Fisher Themis $Z$ double aberration-corrected TEM. The microscope was operated at an accelerating voltage of 300 kV. The images were acquired using a beam current of 10 pA, a convergence angle of 16 mrad and a dwell time of 8 μs. iDPC and ADF images were obtained simultaneously. The ADF detector was set at a collection angle of 25–153 mrad. The iDPC images were formed using a segmented ADF detector. A high-pass filter was applied to the iDPC images to reduce low-frequency contrast. The lattice averaged potential maps were obtained by crystallographic image processing using the software CRISP[50].

**Stability tests and additional characterization**. Stability tests at various pH were carried out by immersing 10 mg of BSS-PB in 1 ml of aqueous solution prepared from deionized (DI) water and an amount of HCl stock solution (0.1 M or 0.1 mM) required to reach the desired pH. The suspensions were prepared in 5 mL Pyrex tubes, sealed, and stirred under ambient conditions for 1 h, after which they were centrifuged at 6000 × $g$ for 5 min. The solids were then washed with 20 ml of DI water, centrifuged again (6000 × $g$ for 5 min), and left to dry in ambient conditions prior to PXRD measurements. In-house PXRD measurements were carried out using a Malvern Panalytical X'pert Pro diffractometer (Cu K$\alpha_{1,2}$, $\lambda_1 = 1.540598$ Å, $\lambda_2 = 1.544426$ Å) using a Bragg−Brentano geometry. Thermogravimetric analysis data were gathered using a TA Instruments Discovery TGA. Scanning electron microscopy images were collected on a JEOL JSM7401F SEM. Calculated (%) for C$_7$H$_5$BiO$_4$ (BSS): C 23.22 H 1.39; measured for BSS-SA (%): C 22.74 H 1.40.

## Data availability

CCDC 2095448 and 2111213 contain the supplementary crystallographic data for this paper. These data can be obtained free of charge via www.ccdc.cam.ac.uk/data_request/cif or by emailing data_request@ccdc.cam.ac.uk or by contacting The Cambridge Crystallographic

Data Centre, 12 Union Road, Cambridge CB2 1EZ, U.K.; fax: + 44 1223 336033. The X-ray powder diffraction data used for the structure refinement has been deposited at Zenodo[51].

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

## Acknowledgements

A.K.I. would like to thank Professor Tomislav Friščić for inspirational correspondence, John Inge for proofreading and improving the manuscript, Riuko Inge for providing Pepto-Bismol samples, as well as Dr. Catherine Dejoie and Dr. William Shepard for data collection. Use of the Advanced Photon Source at Argonne National Laboratory was supported by the U.S. Department of Energy, Office of Science, Office of Basic Energy Sciences, under Contract No. DE-AC02-06CH11357. E.S.G. and A.K.I. acknowledge support from the Swedish Foundation for Strategic Research (SSF) and the Knut and Alice Wallenberg Foundation (KAW 2016.0072). T.W. acknowledges support from the Swedish Research Council (VR, 2019-05465).

## Author contributions

E.S.G. prepared the samples, collected and analysed 3DED, HCA, PXRD, SEM, TGA and FT-IR data and performed structure solution and refinement. V.R. prepared samples and performed PXRD and SEM. M.N. prepared and sectioned samples for STEM imaging. T.W. performed STEM analysis and managed TEM experiments. A.K.I. supervised and managed the project. E.S.G., T.W. and A.K.I. wrote the manuscript. All authors provided feedback and discussed the manuscript.

## Funding

## Competing interests

The authors declare no competing interests.
