## [Peer Review File · Nature Communications]

Structure of the active pharmaceutical ingredient bismuth subsalicylateREVIEWER COMMENTS

Reviewer #1 (Remarks to the Author):

This is a very nice paper dealing with the structure of bismuth subsalicylate. Several research groups have been working on this problem for some time and it is nice to see an elegant solution to elucidating the structure. It will make a nice addition to the literature on bismuth subsalicylate.

My only suggestion for the paper is the inclusion of some additional structural reports that possess similar, although not identical, Bi-O networks. All of these are found in the Cambridge database (TIKSIS, UDICAN, KULJEI, LIGJOC). The references are:

Feng, Y.-Q.; Chen, S.-Y.; Wang, L.; Xing, Z.-Z., Ionothermal Synthesis, Structure and Luminescent Properties of a New 2-D Bismuth (III) Coordination Polymer with (6,5-Connected Topological Sheet. Chinese Journal of Structural Chemistry 2018, 37, 1656, DOI: 10.14102/j.cnki.0254-5861.2011-2024.

Tran, D. T.; Chu, D.; Oliver, A. G.; Oliver, S. R. J., A 3-D bismuth–organic framework containing 1-D cationic inorganic [Bi₂O₂]²⁺ chains. Inorganic Chemistry Communications 2009, 12, 1081-1084, DOI: <https://doi.org/10.1016/j.inoche.2009.08.030>.

Wibowo, A. C.; Smith, M. D.; zur Loye, H.-C., New 3D bismuth-oxo coordination polymers containing terephthalate-based ligands: observation of Bi₂O₂-layer and Bi₄O₃-chain motifs. CrystEngComm 2011, 13, 426-429, DOI: 10.1039/C0CE00650E.

Feyand, M.; Köppen, M.; Friedrichs, G.; Stock, N., Bismuth Tri- and Tetraarylcarboxylates: Crystal Structures, In Situ X-ray Diffraction, Intermediates and Luminescence. Chem. - Eur. J. 2013, 19, 12537-12546, DOI: 10.1002/chem.201301139.

Reviewer #2 (Remarks to the Author):

The authors report on the elucidation of the structure of so-called bismuth subsalicylate, which is the active ingredient of a pharmaceutical named Pepto Bismol. Pepto Bismol is an over-the-counter drug in the US with a long history in medical use. It is widely used although that it contains the heavy metal bismuth and is not characterized in full detail. It is indeed surprising that its structure was not fully encoded until now despite several reported attempts and most likely even more unreported attempts. The synthesis is quite simple, but as the authors demonstrate in their report severe disorder hampers structure elucidation by routinely performed single crystal x-ray diffraction. Therefore they have used a combination of sophisticated techniques based on three-dimensional electron diffraction (3DED) and high resolution scanning transmission electron microscopy (STEM) to provide convincing evidence for their conclusions. The authors compared five samples from different suppliers and formulations by PXRD and have chosen a sample from Sigma-Aldrich and one obtained by isolation from a commercial drug formulation. Finally, they report the structure of so-called bismuth subsalicylate and present a first idea how dissolution of the drug might start.

I am sure that from a chemical point of view the paper is of great interest for a large community, but of course especially for those working in the field of bismuth chemistry in general and its biomedical use in specific. However, it is also an interesting work presenting the use of sophisticated structure elucidation techniques for otherwise unsuccessful SCXRD. Thus, the manuscript is recommended after some minor points were addressed.

Bismuth subsalicylate (BSS) is a trivial name, familiar for those working with such compounds, however the systematic name should also be provided. The term "sub" is often used in other chemical context and thus sometimes misleading for non-specialists.

The authors state that Pepto Bismol was bought from a supermarket. More detailed information on the supplier should be provided.

There is a difference in the PXRDs of the different formulations. A more detailed explanation might be given.

The authors state (line 40) that BSS is “a crystalline compound made from Bi³⁺ and salicylic acid”. This description is misleading. Either the authors report on the synthetic procedure including the starting materials as used or on structural details, which requires the salicylate and oxido ligands instead of salicylic acid. With this regard presentation of salicylic acid in fig. 1 might be misleading as well.

The authors state (line 45) bismuth compounds have been demonstrated to combat antibiotic resistance in bacteria and suppress SARS-COV-2 replication. The compound names might be given and some more examples on antimicrobial compounds of bismuth might be added.

SEM of the sample from Aldrich looks quite different from that of the isolated commercial BSS. Does BSS-SA looks similar after aqueous treatment?

The authors state (line 125) that the IBU of BSS is nearly identical to the one found in a previously reported bismuth-biphenyltricarboxylate coordination polymer. This seems to be incorrect. In the coordination polymer oxido groups are not present and the coordination numbers of bismuth differ as compared to BSS. The similarity is closer to [Bi₄O₂(Hsal)₈]-2MeCN/MeNO₂. If the authors refer to the coordination number and coordination environment in coordination polymers with aryl carboxylates broader citation/discussion is necessary.

The authors use the term phenol group in the text. I recommend to use the term phenolic group.

The authors state (line 135) that “phenol groups (are) still protonated.” It sounds as if deprotonation of the phenolic OH of the salicylate ligand would have been expected, but based on the literature such an assumption is not supported. In line 149 the authors give two examples of phenolic OH groups (for salicylate compounds), however, the list could be extended. On the other hand in [Bi(Hsal)(sal)(H₂O)] both situations are realized. This discussion might be extended.

The authors state (line 145) that “the carboxylate oxygen atoms form relatively shorter Bi-O bonds (2.6–2.8 Å) compared to the phenol oxygen atoms (2.8–2.9 Å).” The text sounds misleading and should be rewritten to be more clearly.

In line 211 the authors start to discuss shortly the dissolution process of BSS. Do they have any idea in which form or formulation the compound will reach the gastric lining. Others have reported that bismuth oxo salicylate clusters play a role. Do the authors think that these ideas are in contradiction to the results reported here? Is a suspension of BSS supposed to reach the stomach lining? What is the pH of the stomach lining and is it supposed to form BiOCl at its place of biomedical effect?

The authors state (line 469) the absence of a C=O stretching vibration in the IR spectrum and refer to the salicylic acid. However, even in carboxylates C=O stretching vibrations are present, but their frequencies are shifted as a result of different bond strength, bond character and coordination mode.

Reviewer #3 (Remarks to the Author):

The authors determined the structure of bismuth subsalicylate, a widely used API molecule of significant clinical importance. This compound has a very long history, but its structure has remained elusive due to its propensity for disorder as elegantly described in this paper. EDED was used to determine the structure of bismuth subsalicylate as prepared by Sigma Aldrich. Pharmaceutical formulations were shown to be composed of layers of the structure determined by 3DED by STEM with several disordered arrangements. Additional analysis of the bulk formulation by PXRD was carried out. The authors conclude that the carboxylic acid and not the phenol is deprotonated in the crystal lattice and support this finding with IR spectroscopy. Together these findings highlight a highly appropriate application of advanced TEM methods: 3DED and STEM, and further our understanding of this pharmaceutically important bismuth compound.

This paper merits acceptance to the journal Nature Communications following minor revisions to the text and addressing some concerns regarding the 3DED structure, concerns that are unlikely to alter the scientific conclusions.

3DED Structure Concerns:

- The completeness of the uploaded CIF file is only 40.7% while Ex Table 1 lists the completeness as being 84.5%. This makes it difficult to evaluate the quality of the structure and an updated CIF file with more complete data is requested.
- There are a large number of DFIX restraints, especially for a data reported to extend to such high-resolution. It is unclear if these are necessary or simply due to refinement against such low completeness data.
- The case for deprotonation of the carboxylic acid group and not the phenol is strengthened by an analysis of bond lengths, not only of the Bi-O bonds, but also the C-O bonds. I do feel that the use of DIFX restraints for these bonds needs to be acknowledged in some way (assuming these are still needed after addressing the completeness issue). Reporting the unrestrained bond lengths in the SI would be appropriate if DFIX restraints are still necessary. Thankfully, the IR data supports the 3DED model, which makes this point less concerning.

Other minor points:

- Please add a resolution ring or other indicator to Figures 1e and Ex Data Fig 4, similar to what was done in Ex Data Fig 3.
- The CCDC number listed for the 3DED structure does not appear to be correct in the text.
- Ex Data Fig 7. Consider re-phrasing the sentence "Note the absence of the C=O stretching vibration of the hydrogen-bonded carboxylic acid groups (1654 cm⁻¹) in the BSS-SA spectra." In opening the crystal structures, it is clear that there are likely hydrogen bonds in this structure, but this network is not discussed in the paper as far as I could tell. Consider rephrasing to "the deprotonated carboxylic acid groups" or something similar to avoid confusion.
- Ex Data Table 1. It is difficult to evaluate the quality and the appropriateness of the resolution cutoff used for these data. Please add CC1/2 and I/sigI to this table. Breaking up these values by resolution shells (e.g. high, low and overall or overall and high) would also be helpful.
- Please address all Level A alerts in the CIF file for the 3DED structure.

Answers to Reviewers' Comments (point-by-point):

Reviewer #1 (Remarks to the Author):

This is a very nice paper dealing with the structure of bismuth subsalicylate. Several research groups have been working on this problem for some time and it is nice to see an elegant solution to elucidating the structure. It will make a nice addition to the literature on bismuth subsalicylate.

My only suggestion for the paper is the inclusion of some additional structural reports that possess similar, although not identical, Bi-O networks. All of these are found in the Cambridge database (TIKSIS, UDICAN, KULJEI, LIGJOC). The references are:

Feng, Y.-Q.; Chen, S.-Y.; Wang, L.; Xing, Z.-Z., Ionothermal Synthesis, Structure and Luminescent Properties of a New 2-D Bismuth (III) Coordination Polymer with (6,5-Connected Topological Sheet. Chinese Journal of Structural Chemistry 2018, 37, 1656, DOI: 10.14102/j.cnki.0254-5861.2011-2024.

Tran, D. T.; Chu, D.; Oliver, A. G.; Oliver, S. R. J., A 3-D bismuth-organic framework containing 1-D cationic inorganic [Bi₂O₂]²⁺ chains. Inorganic Chemistry Communications 2009, 12, 1081-1084, DOI: <https://doi.org/10.1016/j.inoche.2009.08.030>.

Wibowo, A. C.; Smith, M. D.; zur Loye, H.-C., New 3D bismuth-oxo coordination polymers containing terephthalate-based ligands: observation of Bi₂O₂-layer and Bi₄O₃-chain motifs. CrystEngComm 2011, 13, 426-429, DOI: 10.1039/C0CE00650E.

Feyand, M.; Köppen, M.; Friedrichs, G.; Stock, N., Bismuth Tri- and Tetraarylcarboxylates: Crystal Structures, In Situ X-ray Diffraction, Intermediates and Luminescence. Chem. - Eur. J. 2013, 19, 12537-12546, DOI: 10.1002/chem.201301139.

We thank the reviewer for the very insightful comment. Indeed, the four structures mentioned by the reviewer have very similar inorganic building units (IBUs) to that of BSS. The four articles mentioned above have been added to the list of references (reference number 38-41), and their citations added to the discussion of the IBU. The sentence mentioning the similarity of the IBU has been changed to the following (line 132-133):

“The IBU of BSS is very similar but not identical to those found in a number of previously reported bismuth-based coordination polymers^{33,38-41}.”

Reviewer #2 (Remarks to the Author):

The authors report on the elucidation of the structure of so-called bismuth subsalicylate, which is the active ingredient of a pharmaceutical named Pepto Bismol. Pepto Bismol is an over-the-counter drug in the US with a long history in medical use. It is widely used although that it contains the heavy metal bismuth and is not characterized in full detail. It is indeed surprising that its structure was not fully encoded until now despite several reported attempts and most likely even more unreported attempts. The synthesis is quite simple, but as the authors demonstrate in their report severe disorder hampers structure elucidation by routinely performed single crystal x-ray diffraction. Therefore they have used a combination of sophisticated techniques based on three-dimensional electron diffraction (3DED) and high resolution scanning transmission electron microscopy (STEM) to provide convincing evidence for their conclusions. The authors compared five samples from different suppliers and formulations by PXRD and have chosen a sample from Sigma-Aldrich and one obtained by isolation from a commercial drug formulation. Finally, they report the structure of so-called bismuth subsalicylate and present a first idea how dissolution of the drug might start. I am sure that from a chemical point of view the paper is of great interest for a large community, but of course especially for those working in the field of bismuth chemistry in general and its biomedical use in specific. However, it is also an

interesting work presenting the use of sophisticated structure elucidation techniques for otherwise unsuccessful SCXRD. Thus, the manuscript is recommended after some minor points were addressed.

1. Bismuth subsalicylate (BSS) is a trivial name, familiar for those working with such compounds, however the systematic name should also be provided. The term “sub” is often used in other chemical context and thus sometimes misleading for non-specialists.

We agree with the reviewer that the term “sub” has different meanings in chemistry, which could potentially cause confusion, and that the way it is used in “bismuth subsalicylate” is rather outdated.

Based on the 2005 IUPAC Red Book recommendations for the nomenclature of inorganic chemistry, and specifically Table IR-1.6, we believe the title compound should be classified under the category of “ions” described in section IR-5.4 “Generalized stoichiometric names”. Based on the recommendations we suggest the systematic name “bismuth(3+) oxide salicylate”. The following sentence has been added to the manuscript, following the description of the structure (line 127-129):

“Considering the stoichiometry of the material and the presence of O^{2-} anions, the systematic name of BSS is bismuth(3+) oxide salicylate.”

However, “bismuth subsalicylate” and its abbreviation “BSS” are widely used and most recognizable. “Bismuth subsalicylate” is the term that is used on Pepto-Bismol containers, Wikipedia, chemical and pharmaceutical databases and chemical suppliers. Therefore, we would prefer to use the term “bismuth subsalicylate” throughout most of the manuscript.

2. The authors state that Pepto Bismol was bought from a supermarket. More detailed information on the supplier should be provided.

Details of the supplier have now been added to the manuscript (lines 229-230).

“Commercially available Pepto-Bismol suspension manufactured by P&G was purchased from the supermarket Foodland Pukalani, HI, USA.”

3. There is a difference in the PXRDs of the different formulations. A more detailed explanation might be given.

The caplet and chewable tablet forms of Pepto-Bismol contain crystalline inactive ingredients such as calcium carbonate among others, which contribute peaks to the powder patterns. For example, the peak at $2\theta = 29^\circ$ is attributed to calcite. The following sentence has been modified to include this detail (Supplementary Figure 1 caption).

“Pepto-Bismol in chewable tablet and caplet formulations consist of additional crystalline phases such as calcium carbonate as inactive ingredients, which contribute additional reflections.”

However even among the samples that we suspect are pure bismuth subsalicylate (Lab-made, BSS-PB and BSS-SA) there are subtle but real differences to the powder patterns which we believe originate from the different degrees of disorder within the materials. The following sentence has been added to address this within the main manuscript (lines 84-86):

“The differences observed in the powder patterns are attributed to either the presence of crystalline inactive ingredients or various degrees of disorder within the materials.”

4. The authors state (line 40) that BSS is “a crystalline compound made from Bi^{3+} and salicylic acid”. This description is misleading. Either the authors report on the synthetic procedure including the starting materials as used or on structural details, which requires the salicylate and oxido ligands

instead of salicylic acid. With this regard presentation of salicylic acid in fig. 1 might be misleading as well.

The sentence mentioned has been revised to list the contents of the material (line 40-41):

“a crystalline compound made from Bi^{3+} cations, O^{2-} anions, and salicylate anions (Hsal^- , Figure 1a).”

Additionally, Figure 1 has been edited to show a singly deprotonated salicylate anion rather than a salicylic acid molecule, with the corresponding changes to the caption of Figure 1a:

“a Molecular structure of the salicylate anion (Hsal^-).”

5. The authors state (line 45) bismuth compounds have been demonstrated to combat antibiotic resistance in bacteria and suppress SARS-COV-2 replication. The compound names might be given and some more examples on antimicrobial compounds of bismuth might be added.

The aforementioned sentence has been changed to be more inclusive of other commonly used bismuth compounds (lines 44-48):

“Antimicrobial properties of several other bismuth compounds such as bismuth subgallate, bismuth subnitrate, and colloidal bismuth tartrate have been reported⁸, and specifically, it has been demonstrated that colloidal bismuth subcitrate and ranitidine bismuth citrate can even combat antibiotic resistance in bacteria⁹ and suppress SARS-CoV-2 replication¹⁰.”

6. SEM of the sample from Aldrich looks quite different from that of the isolated commercial BSS. Does BSS-SA looks similar after aqueous treatment?

We have now acquired SEM images of washed BSS, as acquired from Sigma-Aldrich. The image has been added as part of Supplementary Figure 2 and shows no difference from that of the as-received sample before washing.

The caption of Supplementary Figure 2 has been updated accordingly:

“b) Commercially available bismuth subsalicylate from Sigma-Aldrich after washing with water, showing no significant changes to crystal size or morphology. c) SEM image of a Pepto-Bismol suspension (BSS-PB) after washing with water.”

7. The authors state (line 125) that the IBU of BSS is nearly identical to the one found in a previously reported bismuth-biphenyltricarboxylate coordination polymer. This seems to be incorrect. In the coordination polymer oxido groups are not present and the coordination numbers of bismuth differ as compared to BSS. The similarity is closer to $[\text{Bi}_4\text{O}_2(\text{Hsal})_8] \cdot 2\text{MeCN}/\text{MeNO}_2$. If the authors refer to the coordination number and coordination environment in coordination polymers with aryl carboxylates broader citation/discussion is necessary.

The figure below might help illustrate the similarities of the two compounds. For all three structures only bismuth cations as well as oxido anions (O^{2-}) are shown for clarity.

BSS and the bismuth-biphenyltricarboxylate coordination polymer $\text{Bi}_2\text{O}_2(\text{HBPT})$ (CCDC 1926729, VUVXIW) both consist of infinite rod-shaped IBUs of bismuth cations and oxido anions coordinated in a μ_3 and μ_4 manner. On the other hand, the structure published by Thurston *et al.* consists of a tetranuclear bismuth cluster.

However, thanks to this comment and comments from reviewer #1, a number of additional structures possessing very similar IBUs have been found and added together with their references (references 38-41). The sentence describing the similarity has been changed to the following (lines 132-133):

“The IBU of BSS is very similar but not identical to those found in a number of previously reported bismuth-based coordination polymers^{33,38-41}.”

Even so, when taking into account the Hsal^- ligands, as the reviewer points out, we do recognize some similarities in BSS and the $[\text{Bi}_4\text{O}_2(\text{Hsal})_8] \cdot 2\text{MeCN}/\text{MeNO}_2$ structures in how some of the Hsal^- ligands coordinate to Bi^{3+} cations and bridge neighbouring IBUs (see figure below). The following was added (lines 137-138):

“the phenolic group coordinates to Bi^{3+} cations in adjacent IBUs, similarly to the bridging Hsal⁻ ligands in the $[\text{Bi}_4\text{O}_2(\text{Hsal})_8] \cdot 2\text{MeCN}/\text{MeNO}_2$ structures,¹⁶”

8. The authors use the term phenol group in the text. I recommend to use the term phenolic group.

All occurrences of “phenol group” have been replaced by “phenolic group” throughout the manuscript.

9. The authors state (line 135) that “phenol groups (are) still protonated.” It sounds as if deprotonation of the phenolic OH of the salicylate ligand would have been expected, but based on the literature such an assumption is not supported. In line 149 the authors give two examples of phenolic OH groups (for salicylate compounds), however, the list could be extended. On the other hand in $[\text{Bi}(\text{Hsal})(\text{sal})(\text{H}_2\text{O})]$ both situations are realized. This discussion might be extended.

As the reviewer explains, the carboxylate is more likely to be deprotonated than the phenolic oxygen atoms, which is due to the higher acidity of carboxylic acid groups compared to phenolic groups. We have rephrased the sentence as follows (lines 149-153) in an attempt to make it clearer, and have left out discussions about the protonation of the phenolic groups to prevent confusion:

“Deprotonation of the carboxylate groups is suggested by the 3DED data based on the generally shorter Bi-O(carboxylate) distances (2.3–3.1 Å) compared to the Bi-O(phenolic) distances (3.1–3.2 Å) (Supplementary Table 3). IR spectroscopy also indicates full deprotonation of the carboxylate groups (Supplementary Figure 7). The protonation, as assigned in Figure 2d, results in a charge-balanced material.”

10. The authors state (line 145) that “the carboxylate oxygen atoms form relatively shorter Bi-O bonds (2.6–2.8 Å) compared to the phenol oxygen atoms (2.8–2.9 Å).” The text sounds misleading and should be rewritten to be more clearly.

As aforementioned in the response to the previous comment, the sentence (now lines 149-151) has been modified to clarify the different types of Bi-O distances being compared. The values of the Bi-O distances have also been updated after removing the DFIX distance restraints on all Bi-O distances, which resulted in a broader range of Bi-O(carboxylate) distances and longer Bi-O(phenolic) distances, which are now listed in Supplementary Table 3.

11. In line 211 the authors start to discuss shortly the dissolution process of BSS. Do they have any idea in which form or formulation the compound will reach the gastric lining. Others have reported that bismuth oxo salicylate clusters play a role. Do the authors think that these ideas are in contradiction to the results reported here? Is a suspension of BSS supposed to reach the stomach lining? What is the pH of the stomach lining and is it supposed to form BiOCl at its place of biomedical effect?

Considering that the acidity of the stomach fluctuates from roughly pH 1 – 3, BSS is expected to only be partially converted into bismuth oxychloride upon reaching the stomach. As stated by Bierer (reference 1 in main text, full reference below) and DuPont *et al.* (reference 43 in main text, full reference below), BSS likely sequentially converts to bismuth oxychloride and bismuth oxycarbonate, which are then further transformed to bismuth sulfide throughout the gastrointestinal tract. As such, we believe that the dominant species in the stomach are BSS and bismuth oxychloride, with ratios depending on the pH of the stomach. As such, either of the two species are likely to reach the gastric lining, where the hydrophobic surface of the BSS crystals could be favorable in restoring a normal gastric environment, as highlighted by Goggin *et al.* (reference 44 in main text, full reference below). We have added a reference to DuPont *et al.* (reference 43 in main text, full reference below) and added the following part to the following sentence (lines 213-215:

“At a pH of 2, a small proportion of BSS converted into bismuth oxychloride, and was fully converted at a pH of 1, which is in line with previous indications that some of the administered BSS could even reach the small intestine^{1,43}.”

1. Bierer, D. W. Bismuth Subsalicylate : History , Chemistry , and Safety Author (s): Douglas Ws . Bierer Source : Reviews of Infectious Diseases , Vol . 12 , Supplement 1 . Pathophysiology of Gastrointestinal Infections : The Role of Bismuth Subsalicylate (Jan . – Feb. *Oxford Univ. Press* **12**, (1990).
43. DuPont, H. L. Bismuth Subsalicylate in the Treatment and Prevention of Diarrheal Disease. *Drug Intell Clin Pharm* **21**, 687–693 (1987).
44. Goggin, P. M. *et al.* Surface hydrophobicity of gastric mucosa in *Helicobacter pylori* infection: Effect of clearance and eradication. *Gastroenterology* **103**, 1486–1490 (1992).

12. The authors state (line 469) the absence of a C=O stretching vibration in the IR spectrum and refer to the salicylic acid. However, even in carboxylates C=O stretching vibrations are present, but their frequencies are shifted as a result of different bond strength, bond character and coordination mode.

The last part of the caption for Supplementary Figure 7 has been revised to the following sentence, which states the observation more clearly:

“Note the absence of the C=O stretching vibration expected for a protonated carboxylic acid group (1654 cm^{-1}) in the BSS-SA spectra, instead showing a band shifted to lower wavenumber, as expected for a carboxylate ion ($1600 - 1560\text{ cm}^{-1}$).”

Reviewer #3 (Remarks to the Author):

The authors determined the structure of bismuth subsalicylate, a widely used API molecule of significant clinical importance. This compound has a very long history, but its structure has remained elusive due to its propensity for disorder as elegantly described in this paper. EDED was used to determine the structure of bismuth subsalicylate as prepared by Sigma Aldrich. Pharmaceutical formulations were shown to be composed of layers of the structure determined by 3DED by STEM with several disordered arrangements. Additional analysis of the bulk formulation by PXRD was carried out. The authors conclude that the carboxylic acid and not the phenol is deprotonated in the crystal lattice and support this finding with IR spectroscopy. Together these findings highlight a highly appropriate application of advanced TEM methods: 3DED and STEM, and further our understanding of this pharmaceutically important bismuth compound.

This paper merits acceptance to the journal Nature Communications following minor revisions to the text and addressing some concerns regarding the 3DED structure, concerns that are unlikely to alter the scientific conclusions.

1. 3DED Structure Concerns: The completeness of the uploaded CIF file is only 40.7% while Ex Table 1 lists the completeness as being 84.5%. This makes it difficult to evaluate the quality of the structure and an updated CIF file with more complete data is requested.

We thank the reviewer for their insightful comments and for noticing our mistake. A CIF based on the aforementioned 12 merged datasets, with a completeness of 84.5 %, has been now correctly uploaded to the CCDC (and is included as part of the supplementary material). We apologize for the mistake of uploading the incorrect CIF file containing only one of the datasets rather than the merged dataset. CCDC 2111213 now contains the correct CIF, as is given in the main manuscript.

2. There are a large number of DFIX restraints, especially for a data reported to extend to such high-resolution. It is unclear if these are necessary or simply due to refinement against such low completeness data. The case for deprotonation of the carboxylic acid group and not the phenol is strengthened by an analysis of bond lengths, not only of the Bi-O bonds, but also the C-O bonds. I do feel that the use of DFIX restraints for these bonds needs to be acknowledged in some way (assuming these are still needed after addressing the completeness issue). Reporting the unrestrained bond lengths in the SI would be appropriate if DFIX restraints are still necessary. Thankfully, the IR data supports the 3DED model, which makes this point less concerning.

We thank the reviewer for pointing this out. In the updated CIF of CCDC 2111213, all DFIX restraints pertaining to Bi-O distances have been removed during refinement against the merged 3DED data that resulted in the now correct CIF. Removing the DFIX restraints resulted in a broader range of Bi-O(carboxylate) distances, while all Bi-O(phenolic) distances remained very long. Generally, the Bi-O(phenolic) distances are noticeably longer than the Bi-O(carboxylate) distances. It should however be noted that the coordination modes do vary as some carboxylate oxygens coordinate to single Bi

cations (which tend to result in short bonds), while other carboxylate oxygen atoms bridge two bismuth cations (which tend to result in longer bonds). As Reviewer #3 points out, the IR data supports the conclusion that the carboxylates are deprotonated. Also, as Reviewer #2 points out, the carboxylates are expected to be protonated rather than the phenolic oxygens based on literature. This is due to the higher acidity of carboxylic acids than phenolic groups. Taking into account the IR data, bond distances, coordination geometries, charge balance, and the relative acidity of the functional groups, we believe the assigned protonation is both reasonable and consistent. Below is a table of all Bi-O distances below 3.3 Å in the final model refined against the merged 3DED data without DFIX on Bi-O distances. This table has also been added as Supplementary Table 3.

Atom 1	Atom 2	Distance (Å)	Oxygen species
Bi2	O5	2.28	carboxylate
Bi4	O11	2.50	carboxylate
Bi4	O14	2.65	carboxylate
Bi2	O8	2.72	carboxylate
Bi4	O15	2.79	carboxylate
Bi1	O12	2.80	carboxylate
Bi3	O12	2.88	carboxylate
Bi1	O8	2.95	carboxylate
Bi3	O14	2.97	carboxylate
Bi2	O9	3.01	carboxylate
Bi3	O6	3.10	carboxylate
Bi1	O6	3.14	carboxylate
Bi2	O10	3.13	phenolic
Bi4	O16	3.17	phenolic
Bi4	O10	3.23	phenolic
Bi4	O4	2.19	μ ₃ -O
Bi3	O4	2.19	μ ₃ -O
Bi1	O1	2.22	μ ₃ -O
Bi3	O1	2.24	μ ₃ -O
Bi4	O2	2.25	μ ₃ -O
Bi1	O2	2.26	μ ₃ -O
Bi2	O2	2.36	μ ₃ -O
Bi3	O1	2.39	μ ₃ -O
Bi2	O4	2.65	μ ₃ -O
Bi1	O3	2.33	μ ₄ -O
Bi2	O3	2.37	μ ₄ -O
Bi1	O3	2.45	μ ₄ -O
Bi3	O3	2.54	μ ₄ -O

Distances	Ave. (Å)	Min. (Å)	Max. (Å)
Bi-O(carboxylate)	2.81	2.28	3.14
Bi-O(phenolic)	3.18	3.13	3.23
Bi-O(μ_3)	2.30	2.19	2.65
Bi-O(μ_4)	2.42	2.33	2.54

3. Please add a **resolution ring** or other indicator to Figures 1e and Ex Data Fig 4, similar to what was done in Ex Data Fig 3.

Indicators of the resolution has been added to Figure 1e and Supplementary Figure 4, in a similar manner as for Supplementary Figure 3. Their captions have been updated accordingly.

4. The CCDC number listed for the 3DED structure does not appear to be correct in the text.

CCDC 2111213 does correspond to the BSS structure as refined against the collected 3DED data, which has now been updated to be associated with the correct CIF.

5. Ex Data Fig 7. Consider re-phrasing the sentence “Note the absence of the C=O stretching vibration of the hydrogen-bonded carboxylic acid groups (1654 cm⁻¹) in the BSS-SA spectra.” In opening the crystal structures, it is clear that there are likely hydrogen bonds in this structure, but this network is not discussed in the paper as far as I could tell. Consider rephrasing to “the deprotonated carboxylic acid groups” or something similar to avoid confusion.

The caption of Supplementary Figure 7 has been updated to end with the following sentence:

“Note the absence of the C=O stretching vibration expected for a protonated carboxylic acid group (1654 cm⁻¹) in the BSS-SA spectra, instead showing a band shifted to lower wavenumber, as expected for a carboxylate ion (1600 - 1560 cm⁻¹).”

6. Ex Data Table 1. It is difficult to evaluate the quality and the appropriateness of the resolution cutoff used for these data. Please add CC1/2 and I/sigI to this table. Breaking up these values by resolution shells (e.g. high, low and overall or overall and high) would also be helpful.

Two rows have been added in Supplementary Table 1, showing I/sigI for two resolution ranges (Inf. – 0.8 Å, 0.9 – 0.8 Å), as well as CC1/2 for the merged data.

7. Please address all Level A alerts in the CIF file for the 3DED structure.

Comments have been added for all the A-level and B-level alerts encountered in the 3DED CIF.

REVIEWER COMMENTS

Reviewer #2 (Remarks to the Author):

The authors have addressed the reviewers concerns thoroughly. I would like to recommend publication of this nice work in its current form.

Reviewer #3 (Remarks to the Author):

The authors have adequately addressed the reviewer concerns and I recommend this article for publication.